# The Impact of Corporate Social Responsibility on Environmental Investment: The Mediating Effects of Information Transmission and Resource Acquisition

**Ruizhi Liu** [1] , **Fei Song** [1] , **Mark Wu** [2] **and Yuming Zhang** [3,*]

1   Management College, Ocean University of China, Qingdao 266100, China; 13370812971@163.com (R.L.); songfouc@163.com (F.S.)
2   Mario J. Gabelli School of Business, Roger Williams University, One Old Ferry Road, Bristol, RI 02809, USA
3   School of Management, Shandong University, Jinan 250100, China
*   Correspondence: zhangym@sdu.edu.cn

**Abstract:** In recent years, more and more research has focused on the impact of corporate social responsibility (CSR) on business activities. Due to the existence of two different theoretical perspectives, shareholder value theory and managerial opportunism theory, the research on CSR has reached different conclusions. Meanwhile, the motivations for environmental investments in enterprises have received attention from scholars. However, there is a lack of empirical research on the relationship between CSR and environmental investment. Therefore, this study conducts a regression analysis on the external evaluation of CSR and enterprises' environmental investment using data from Chinese listed companies. The empirical results show a significant positive relationship between the external evaluation of CSR and enterprises' environmental investment. The mediating tests conducted based on information transmission and resource acquisition mechanisms explain the reasons for this promotion effect, supporting the shareholder value theory. Furthermore, our research finds that this promotion effect is more significant in non-state-owned enterprises, enterprises receiving fewer environmental subsidies, enterprises disclosing environmental philosophies, and enterprises identified as key pollution-monitoring units in reports. The research findings of this study are meaningful for clarifying the economic consequences of CSR and provide practical evidence for Chinese enterprises to understand the importance of environmental investment and the government's advocacy for enterprises to proactively engage in environmental investment.

**Keywords:** corporate social responsibility; environmental investment; shareholder value theory

## 1. Introduction

Environmental investment in enterprises refers to the deliberate allocation of resources towards promoting greener production processes and gaining a competitive advantage in terms of environmental sustainability [1]. It is a proactive measure taken to address existing or potential environmental issues and to achieve a harmonious relationship between human activities and the environment. By embracing the principles of sustainable development and the construction of ecological civilization, environmental investment contributes to the long-term economic and social well-being of society. In essence, environmental investment represents the additional costs incurred by enterprises to mitigate environmental pollution and address environmental challenges. Consequently, the benefits derived from such investments are predominantly social and environmental in nature, rather than purely economic [2].

Enterprises, as key stakeholders in environmental governance, bear the responsibility and obligation to actively engage in pollution prevention and control measures resulting from their business operations. They should strive to mitigate the ecological damage caused

by their management activities. Economist Michael Porter introduced the Porter Hypothesis [3], which posits that environmental protection and economic development are not necessarily conflicting objectives. He contended that although environmental regulations might initially increase a firm's costs due to the need for innovation and investment in environmental protection, they can confer long-term advantages. By fostering resource efficiency, adhering to government environmental policies, minimizing uncertainties in environmental investment, and elevating their environmental consciousness, firms can accrue substantial benefits. Such a strategy leads to an increase in environmental patents [4], affirming the economic value derived from endeavors in environmental protection. This can help offset the costs associated with environmental investment, leading to a "win–win" scenario of environmental governance and profitability [3,5]. However, in the short term, environmental investment often results in environmental and social benefits outweighing economic benefits [6]. Limited financial resources allocated to environmental protection by enterprises can increase production costs [7], reduce production efficiency [8], and diminish their willingness to invest in environmental protection [7–9]. Entrepreneurs driven by immediate profits and economic benefits may hesitate to allocate limited financial resources to short-term environmental investments, which offer low economic returns and uncertain future outcomes [9,10]. Additionally, current environmental expenditures have not yet produced a significant impact on pollution reduction [11]. Therefore, it is crucial to guide and motivate enterprises to actively invest in environmental protection, as it is vital for pollution prevention and control, ecological civilization construction, and the promotion of green development principles.

Consequently, scholars have made significant progress in exploring the factors influencing environmental investment in enterprises. Existing literature primarily focuses on individual perspectives of enterprises when examining motivations for environmental investment [12,13]. Scholars have also delved into the micro-mechanisms behind the actual impact of government inspections on environmental investment in enterprises, providing valuable insights for environmental regulators [14]. Additionally, research suggests that the EU Emissions Trading System serves as a feasible tool and a key incentive for enterprises to bear environmental expenditures [15]. The impact of internet penetration on government environmental protection expenditures has been explored, shedding light on various environmental policy issues [16]. Furthermore, studies have analyzed the influence of CEO beliefs on corporate sustainability expenditure, offering implications for sustainability literature and practices [17]. The nature and extent of strategic interactions in environmental expenditures among OECD countries have also been discussed, highlighting the significant positive spatial dependence that suggests neighboring countries' behavior influences the choice of environmental expenditure policies [18]. Moreover, enterprises should adhere to the principle of minimizing the cost of implementing environmental protection measures when making environmental investment decisions. This includes considering the cost of individual environmental facilities, local and industry-specific environmental regulations, and conducting environmental economic analysis [19]. Additionally, the cost of schemes in micro-level environmental decision-making should be taken into account [20]. Under specific political–economic systems, national institutions [12], government capacity [21], and environmental regulations [22–25] all significantly impact environmental investment in enterprises and cannot be ignored. Despite the progress made in researching environmental investment in enterprises, there is still room for improvement in both the benefits and effects of such investments under various government environmental regulations worldwide. Moreover, the existing literature has yet to thoroughly and comprehensively explore the mechanisms that incentivize and constrain environmental investment in enterprises.

Corporate social responsibility plays a crucial role in aligning economic and social development, ecological civilization construction, and social harmony and stability. As found in historical literature, British scholar Oliver Sheldon is credited with introducing the idea [26]. Sheldon proposed that business pursuits should extend beyond maximizing shareholder profit to encompass basic economic duties, legal compliance, service quality

improvement, community engagement, and broader societal contributions. The concept received further refinement by Howard R. Bowen [27]. Here, Bowen articulated CSR's definition as the commitment of business leaders to harmonizing their policies and decisions with societal needs and values, and to undertake actions that have positive repercussions on economic and social progress, environmental sustainability, and societal well-being. The issuance of third-party social responsibility evaluations can have both incentive effects, encouraging enterprises to actively engage in environmental investment, and concealment effects, leading to a reduction in or cessation of environmental investment activities following the receipt of favorable evaluations. This, in turn, weakens the environmental governance outcomes of social responsibility evaluations [1]. Enterprises with positive social responsibility evaluations can attract investors [28] and increase their likelihood of engaging in outward investments due to their social responsibility advantages [29]. However, existing literature has yet to provide empirical evidence on whether the release of third-party social responsibility evaluations actually produces environmental governance effects and achieves the objective of "evaluation for improvement." Understanding the impact and potential mechanisms of social responsibility evaluations on corporate environmental investment is of paramount theoretical significance and practical value. It can aid in fully harnessing the soft regulatory role of social responsibility evaluations.

Scholars generally support two distinct perspectives on the economic implications of corporate social responsibility: the "shareholder value theory" and "managerial opportunism" perspectives. The "shareholder value theory" argues that embracing social responsibility can benefit companies by providing strategic resources and enhancing corporate value [30]. In contrast, the "managerial opportunism" perspective suggests that fulfilling social responsibility is merely a superficial or cosmetic behavior by management. According to this view, the more social responsibility a company assumes, the more severe the agency problem becomes, with no positive impact on operational performance [31]. The controversy surrounding the economic consequences of corporate social responsibility can be attributed to several factors. Firstly, financial performance is often influenced by non-operational factors and may not accurately reflect a company's competitive advantages [32]. Moreover, there are various methods for measuring financial performance, resulting in diverse relationships between social responsibility and financial performance. Consequently, it becomes challenging to definitively determine the true impact of social responsibility from a purely financial perspective. Secondly, research from a stakeholder perspective highlights that different individuals may possess different prior beliefs or interpret the same information differently [33], and different types of investors may also assess the same information differently [34].

Drawing from existing research, scholars have explored and examined the economic implications of corporate social responsibility from the two theories. Firstly, they have investigated the relationship between corporate social responsibility and financial performance [30,35–39]. Secondly, scholars have focused on the stakeholder perspective, delving into various aspects such as creditors [40], bank risk-taking [41], firm value [42], working capital management [43], investment efficiency [44], dividend policy [45], credit risk [46], stock market [47,48], and government [49] to explore and examine the economic consequences of corporate social responsibility. The literature on the economic consequences of corporate social responsibility presents diverging viewpoints regarding the two perspectives. Different stakeholders may exhibit significantly different reactions to the same corporate social responsibility actions. These apparent differences arise from the opposing and contradictory nature of the shareholder value theory and managerial opportunism. Consequently, it remains challenging to ascertain the genuine economic consequences of corporate social responsibility from the stakeholder perspective.

In recent years, scholars have extensively investigated the motivations behind companies' environmental investments from various dimensions, including the institutional, organizational, and individual managerial levels [12]. However, there is a lack of substantial evidence regarding the specific relationship between social responsibility and

environmental investments. Additionally, most existing research on the motives for environmental investments by companies has primarily focused on the individual firm perspective [12,13]. Similarly, within specific political–economic systems, factors such as national institutions [12], government capabilities [21], and environmental regulations [22–25] significantly influence a company's environmental investments, indicating the need for further examination. Recognizing these research gaps, our study aims to achieve two objectives. The first objective is to analyze the motives for environmental investments by companies and explore their association with social responsibility. This analysis will consider variables such as corporate reputation [50], institutional investor attention [51], operating cash flow [52], and financing constraints [53]. The second objective is to investigate this relationship within the context of developing countries, specifically focusing on China.

The Chinese context presents a captivating backdrop for the study of corporate social responsibility disclosure and its economic implications. Since 1992, China has undergone rapid economic development, resulting in significant environmental issues stemming from high levels of energy consumption. As awareness of environmental protection grows and the public demands a better environment, companies face increasing social pressures to invest in environmental preservation. Concurrently, the government has implemented requirements for companies to disclose social responsibility information, while companies also strive to cultivate a positive public image. China announced a new goal of "striving to reach the carbon peak by 2030 and carbon neutrality by 2060" in September 2020 [54,55]. This target imposes more stringent environmental investment requirements on companies. However, despite these goals, frequent severe environmental incidents still occur, and compliance with PM2.5 standards needs improvement. The performance pressure from the Chinese stock market further compounds the challenges faced by listed companies, possibly leading to their hesitance to make substantial environmental investments. Despite being a China-focused study, these conflicting factors make this topic intriguing and necessitate an exploration of the mechanisms through which corporate social responsibility disclosure influences environmental investments.

This study makes several key contributions: (1) We present empirical evidence that suggests external evaluations of CSR can have a substantial and favorable impact on a firm's environmental investments, aligning with the shareholder value theory. (2) To the best of our knowledge, we believe this is the first analysis to analyze the mediating roles of information transmission and resource acquisition between CSR and environmental investment. The rationale for the information transmission mechanism is the positive "back-pressure" effect generated by social reputation and increased institutional investor interest, while the resource acquisition mechanism is validated by the ample cash flow and reduced financing constraints that enable more resources for environmental investment. Both mechanisms play a crucial role in promoting sustained environmental investment by firms. (3) By conducting a series of heterogeneity tests, our analysis delves into the varying impacts of CSR on environmental investment across different corporate environments. We discover that CSR wields a more pronounced influence on privately-owned firms, firms that receive fewer environmental subsidies, those with more robust environmental disclosure practices, and entities specifically targeted for pollution monitoring. These insights enhance our comprehension of the dynamics that encourage environmental investments and contribute to the theoretical discourse surrounding CSR and its implications.

The rest of the paper structure is as follows: Section 2 introduces relevant literature and proposes the hypothesis of the relationship between CSR performance and environmental protection investment, Section 3 arranges the empirical research design, Section 4 covers the research findings of empirical results and their analysis and discussion, Section 5 provides policy implications and discussions, and Section 6 is the conclusion.

## 2. Literature Review and Development of Hypotheses

### 2.1. Shareholder Value Theory

The theoretical foundation of shareholder value maximization comprises the resource dependency theory and the stakeholder theory. The resource dependency theory refers to an organization's primary objective of reducing dependence on external organizations for critical resource supply and seeking methods to stabilize control over these resources. This theory highlights the need for organizations to extract resources from the surrounding environment and rely on mutual interdependence and interaction to achieve their survival goals. Due to environmental uncertainty and limited resources, organizations may strive for additional resources to safeguard their interests and mitigate the impact of environmental changes, thereby constructing their competitive advantage based on core resources controlled by stakeholders.

On the other hand, based on the stakeholder theory, companies develop social capital by collaborating with transaction partners such as shareholders, creditors, employees, consumers, suppliers, customers, governmental agencies, local residents, local communities, media, and environmental organizations, overcoming the constraints of limited resources through complementary advantages. This theory suggests that companies can be seen as a collection of relationships with various stakeholders, and undertaking social responsibility helps meet stakeholders' expectations and gain access to critical strategic resources held by stakeholders. As the fulfillment of social responsibility deepens, socially responsible companies attract attention from numerous stakeholders, who are more inclined to trust and establish connections with companies demonstrating a high level of social responsibility fulfillment. Similarly, proactive engagement in social responsibility also garners recognition and favor from potential suppliers or customers in the supply chain, leading to more collaboration opportunities and reduced dependence on specific upstream and downstream enterprises. Therefore, under the perspective of shareholder value maximization, the resource aggregation effect of corporate social responsibility brings competitive advantages and enhances corporate value, while also providing a solid basis for proactive investment in environmental protection. Research has shown that fulfilling social responsibility attracts positive media attention [56], contributes to establishing a favorable media image [57], and enhances market reputation [58]. From a supply chain standpoint, companies that actively fulfill social responsibility not only attract responsible consumers, enabling product differentiation strategies and increased profit margins [59], but also experience rapid growth in sales revenue [60], thereby improving the operational status of socially responsible companies and generating positive feedback for environmental investments [61]. Additionally, a company's strong social reputation helps establish trust with suppliers and customers [62], not only attracting attention from external investors for socially responsible companies but also facilitating the acquisition of financial resources from socially responsible investors, thereby improving financing accessibility [63,64] and obtaining the necessary funds for environmental investments.

Based on the aforementioned analysis, external evaluations of corporate social responsibility fully reflect the company's fulfillment of social responsibility, thereby effectively meeting stakeholders' expectations and facilitating access to critical strategic resources held by stakeholders. Therefore, from the perspective of shareholder value maximization, external evaluations of corporate social responsibility attract strategic resources, bringing competitive advantages and enhancing corporate value, thereby reflecting shareholder value. According to the resource dependency theory and the stakeholder theory, the fulfillment of social responsibility by companies is conducive to acquiring strategic resources held by stakeholders such as banks and institutional investors, primarily through establishing a good reputation, maintaining and attracting institutional investments, promoting improvements in business performance, and effectively alleviating financing constraints [65–70]. Based on the discussion above, the following hypothesis is proposed:

**Hypothesis 1a.** *A positive association exists between corporate social responsibility disclosure and environmental investment by companies.*

*2.2. Managerial Opportunism Theory*

Managerial opportunism is grounded in the agency theory, which highlights the inherent conflicts of interest between principals (shareholders) and agents (managers) due to divergent objectives. Research by [71] has indicated that fulfilling social responsibility can lead to decreased profitability and potential harm to shareholder interests, imposing costs on companies. Furthermore, Ref. [72] found that both state-owned and private enterprises experience significant decreases in profitability and shareholder value when assuming social responsibility. This suggests that while fulfilling social responsibility can enhance social welfare, it may come at the expense of shareholder interests. It is important to recognize that actively fulfilling social responsibility can create a positive social reputation and favorable media image for a company, attracting necessary resources. However, it can also be viewed as an agency activity by managers aimed at building personal image and social status [47,73]. With stakeholders placing increasing emphasis on corporate social responsibility, managers face significant pressure to achieve positive outcomes in this domain [74]. For instance, many CEOs of American companies actively pursue social responsibility to satisfy their personal desire for social status or engage in reputation management [75]. In such cases, a company's fulfillment of social responsibility may primarily serve as a means for managers to enhance their social status and personal image, rather than creating value for shareholders. A higher external evaluation of social responsibility reflects a company's proactive approach to fulfilling social responsibility, benefiting managers' interests while potentially harming shareholder interests. The resources and advantages generated during the process of fulfilling social responsibility are controlled by management, while the risks and crises stemming from these activities are borne and shared by shareholders. Consequently, a higher external evaluation of social responsibility exacerbates existing agency problems within a company. Managerial opportunism suggests that the information transmission mechanism resulting from a company's assumption of social responsibility can be exploited by managers to conceal bad reputation or unethical behavior, enhance professional image, and divert public attention [76]. Existing literature on managerial opportunism indicates that assuming social responsibility can lead to negative effects such as concealing managerial earnings management behavior [77], increasing the risk of stock price collapse [47,78], promoting corporate financialization [79–81], and inhibiting bank risk-taking [41]. Based on the discussion above, the following hypothesis is offered:

**Hypothesis 1b.** *A negative association exists between corporate social responsibility disclosure and environmental investment by companies.*

## 3. Model Setting and Data Selection

*3.1. Data Sources*

The main source of the sample in this study is the A-share listed companies on the Shanghai and Shenzhen stock exchanges from 2010 to 2021. We applied the following criteria for sample selection: (1) Excluding financial companies such as banks, insurance, and diversified financial companies; (2) eExcluding companies that had been delisted during the research period; (3) excluding companies marked with abnormal statuses such as ST or *ST in the current year; and (4) excluding samples with undisclosed or incomplete financial data. The Shenzhen Stock Exchange and Shanghai Stock Exchange issued guidelines on corporate social responsibility in 2006 and 2008, respectively, but prior to 2010, very few companies disclosed social responsibility-related information. To ensure sufficient observation samples for each year within the research period, this study started with 2010. To avoid the interference of outliers, this study applied a 1% truncation to all continuous variables. To reduce the possibility of missing data, in the process of collecting data on corporate environmental investment, two groups of researchers

independently conducted manual statistics on the data simultaneously and then cross-checked and organized the collected data. The external evaluation of corporate social responsibility is based on data from Hexun.com. Other data used in this study were sourced from the CSMAR database.

*3.2. Variable Setting*

3.2.1. Dependent Variable

Environmental protection investment (EPI) plays a crucial role in the construction of ecological civilization and sustainable development. In line with [53], this study adopted standardized environmental protection investment (EPI) as the dependent variable. It was derived by standardizing the company's year-end total assets against its environmental protection investment. The original data regarding the scale of environmental protection investment were obtained from the construction-related details in the annual reports of listed companies. These details include expenses directly associated with environmental protection, such as desulfurization projects, denitrification projects, wastewater treatment, exhaust gas treatment, dust removal, and energy conservation measures. The annual increase in environmental protection investment was calculated based on these project data. To account for differences in company size, the environmental protection investment was standardized against the company's year-end total assets. Furthermore, for ease of interpretation in the subsequent analysis, the standardized environmental protection investment was multiplied by 100. In the robustness test conducted later in the analysis, greening fees and pollution fees from the "management expenses" item in the company's income statement were included as an additional proxy variable for environmental protection investment. This was carried out alongside the capitalized environmental protection expenditures mentioned earlier, aiming to test the robustness of the results.

3.2.2. Explanatory Variable

The external evaluation of corporate social responsibility (CSR) was measured based on Hexun.com's assessment of all listed companies' fulfillment of social responsibility. Hexun.com focuses on five key aspects: shareholder responsibility, employee responsibility, supply chain responsibility, environmental responsibility, and social responsibility. Their evaluation methodology involves a comprehensive approach, utilizing secondary and tertiary indicators to assess social responsibility. This includes 13 secondary indicators and 37 tertiary indicators, providing a thorough and detailed assessment. In this study, the social responsibility data from Hexun.com were utilized as independent variables, following the approach of [52,82,83]. These references served as a basis for incorporating the Hexun.com data into the analysis. Furthermore, in the robustness test conducted later in the analysis, 12 aspects from the "Basic Information Form of Corporate Social Responsibility Reports" in the Guotaian database were used to create sub-item dummy variables. These variables were then standardized after aggregation and served as an additional proxy variable for the external evaluation of corporate social responsibility in the robustness test. This was carried out to test the robustness of the results obtained from the use of Hexun.com data.

3.2.3. Mediating Variable

This study investigates the underlying reasons for the promotion of environmental investment through corporate social responsibility (CSR), focusing on the information transmission and resource acquisition mechanisms. Firstly, we suppose that enhanced transparency of corporate information can exert pressure on companies, boost their reputation, and attract institutional investors. This, in turn, motivates companies to maintain a positive social image and make substantial environmental protection investments to foster green innovation and reduce pollution, as required in the era of sustainable development. Secondly, companies require more resources and fewer constraints to allocate a higher budget for environmental protection investments, considering the limited direct economic

benefits they can obtain from such investments. According to the discussion above, we use the following mediating variables to validate these mechanisms.

To examine the information transmission path, we selected two mediating variables: corporate reputation (CR) and institutional investor ownership ratio (IIP). To measure corporate reputation (CR), a reputation evaluation system was constructed, incorporating assessments from various stakeholders. Twelve reputation evaluation indicators were chosen, and factor analysis was employed to calculate the reputation scores. The companies were then divided into 10 groups based on their reputation scores, ranging from 1 to 10, with higher scores indicating a better reputation. The institutional investor ownership ratio (IIP) was calculated following the methodology of reference [51]. It is determined by summing the year-end holdings of all institutional investors and dividing it by the total number of company shares. A higher IIP suggests that institutional investors possess a more comprehensive understanding of the company's information and engage in larger-scale investments, indicating greater transparency of corporate information.

To examine the resource acquisition path, we selected two mediating variables: operating cash flow (OCF) and financial constraints. To calculate operating cash flow (OCF), the approach from reference [82] was followed. The OCF is determined by taking the difference between cash inflows and cash outflows from operating activities and dividing it by the total assets of the company at the end of the period. This metric serves as an indicator of the company's internal resource acquisition capability. A higher OCF indicates a stronger operational capability and a more stable internal resource acquisition capacity. To construct the WW index, which reflects the level of financial constraint faced by the company, reference [53] was consulted. This index utilizes five relevant financial indicators and their corresponding weights, as determined by the existing literature. The WW index provides an assessment of the severity of the financial constraint faced by the company. A higher value of the WW index suggests a more pronounced financial constraint, as illustrated in Equation (1).

$$\text{WW} = -0.091 \times \text{CF}_{i,t} - 0.062 \times \text{DIVPOS}_{i,t} + 0.021 \times \text{LEV}_{i,t} - 0.044 \times \text{SIZE}_{i,t} + 0.102 \times \text{ISG}_{i,t} - 0.035 \times \text{SG}_{i,t} \quad (1)$$

In Equation (1), CF represents the cash flow-to-total assets ratio, which is calculated as the net cash flow from operating activities divided by the total assets, and DIVPOS is a dummy variable that indicates whether cash dividends were paid in the current period. It takes a value of 1 if cash dividends were paid and 0 otherwise. LEV denotes the long-term debt-to-assets ratio; SIZE represents the natural logarithm of total assets; ISG refers to the industry-average sales growth rate, using the two-digit industry code for the manufacturing industry and the one-digit industry code for other industries, based on the 2012 industry classification standards of the China Securities Regulatory Commission; and SG denotes the sales revenue growth rate.

### 3.2.4. Control Variables

This study incorporated nine control variables that have the potential to influence corporate environmental investment, building on previous research on the relationship between social responsibility performance and environmental investment.

The first set of control variables pertains to various financial and operational characteristics of the company: (1) firm size (size), (2) firm value (TobinQA), (3) financial leverage (ALR), (4) corporate cash flow (cash), (5) operating capacity (OC), and (6) business growth ability (BRIR).

The second set of control variables focuses on corporate governance attributes: (1) board size (board) and (2) key pollutant monitoring unit (KPMU).

Lastly, the study took into account the influence of government environmental subsidies, following the approach of [84]. The government environmental subsidies were determined based on the detailed subsidy items disclosed in the annual report's notes. The number of subsidies related to environmental protection was manually compiled using keywords such as "green", "environmental subsidy", "environment", "sustainable"

development", "clean", and "energy-saving". The relative level of environmental subsidies, adjusted for company size, was calculated as the ratio of the environmental subsidy amount to total assets. Due to the limited scale of the data, they are presented as a percentage.

Additionally, a set of dummy variables was used to control for provincial-fixed effects, industry-fixed effects, and time-fixed effects. The specific descriptions of variables in this research are shown in Table 1.

**Table 1.** Variable definition.

| Variable Classification | Variable | Variable Symbol | Variable Description |
|---|---|---|---|
| Dependent variable | Environmental protection investment | EPI | (Corporate environmental protection investment/total assets of the enterprise at the end of the year) × 100 |
| Explanatory variable | Corporate social responsibility | CSR | The total score of social responsibility professional evaluation of listed companies in Hexun.com |
| Mediating variables | Corporate reputation | CR | Reputation scores, ranging from 1 to 10 |
| | Institutional investor ownership ratio | IIP | Year-end holdings of all institutional investors/total number of company shares |
| | Operating cash flow | OCF | Difference between cash inflows and cash outflows from operating activities/total assets |
| | Financial constraint | WW | Assessment of the severity of the financial constraint faced by the company |
| Control variable | Firm size | Size | Take the natural logarithm of the ending total assets |
| | Corporate value | TobinQA | Tobin's Q value = market value/ T = total assets |
| | Financial leverage | ALR | Asset–liability ratio = gross liability/ total assets |
| | Corporate cash flow | Cash | (Net increase in cash and cash equivalents—net cash flows from financing activities)/total assets |
| | Operation capacity | OC | Total asset turnover = operation revenue/ending balance of total assets |
| | Development | BRIR | Increased rate of business revenue = (operating income for the current period-operating income in the same period of last year)/(operating income in the same period of last year) |
| | Board size | Board | Number of board of directors of listed companies |
| | Key pollution monitoring unit | KPMU | Set the dummy variable, disclose the company as the key monitoring unit in the report, assign a value of 1, otherwise 0 |
| | Environmental protection subsidy | EPS | Environmental protection subsidy amount/total assets |
| | Dummy variable | Pro | Province |
| | Dummy variable | Ind | Industry |
| | Dummy variable | Year | Year |

*3.3. Model Setting*

To examine the impact of external evaluation of corporate social responsibility on corporate environmental investment, this study constructed the following model:

$$EPI_{it} = \beta 0 + \beta 1 CSR_{it} + \beta 2 Controls_{it} + Year + Industry + \varepsilon_{it}, \tag{2}$$

where *EPI* represents the level of corporate environmental investment, and *CSR* represents an external evaluation of corporate social responsibility on environmental investment. *Controls* include various financial and operational characteristics of the company (e.g., firm size, firm value, financial leverage, corporate cash flow, operating capacity, and business growth ability), corporate governance attributes (e.g., board size, and key pollutant monitoring unit), and government environmental subsidies. *Year* and *Industry* represent the dummy variables, controlling for temporal as well as industry-specific factors that may affect environmental investment. $\varepsilon$ represents the error term.

The model aims to assess the relationship between external evaluation of corporate social responsibility and corporate environmental investment while controlling for various factors that may influence environmental investment. The coefficients ($\beta 1$) associated with the external evaluation of CSR will indicate the magnitude and direction of its impact on environmental investment, considering the effects of the control variables.

## 4. Analysis of Empirical Results

*4.1. Descriptive Statistics*

Table 2 presents a descriptive statistical analysis of the main variables, including the sample size, minimum, maximum, mean, median, and standard deviation. The results reveal important insights about the variables under study. Regarding corporate environmental investment (EPI), the average value for A-share listed companies is 23.231, with a median value of 0. This suggests a prevalent issue of insufficient environmental investment among these companies, as most of them fall well below the average level of environmental investment. Additionally, the standard deviation of EPI is four times the mean, indicating substantial variations in environmental investment levels across different companies. In terms of the external evaluation of corporate social responsibility (CSR), the average value is 23.499, with a median value of 21.72. These findings indicate that A-share listed companies generally perform well in fulfilling their social responsibilities. However, the standard deviation of CSR is 15.184, highlighting significant differences in social responsibility performance among different companies. These variations create a solid basis for the research conducted in this study. The descriptive statistical results of the company characteristic variables align with existing literature, further validating the reliability and consistency of the analysis. Overall, the descriptive statistics shed light on the distribution and characteristics of the variables studied, providing important insights into the nature of corporate environmental investment and external evaluation of corporate social responsibility among A-share listed companies.

**Table 2.** Descriptive statistics.

| Variable | Minimum Value | Maximum Value | Mean Value | Standard Deviation |
|:---:|:---:|:---:|:---:|:---:|
| EPI | 0.000 | 677.621 | 23.231 | 93.265 |
| CSR | −3.710 | 73.690 | 23.499 | 15.184 |
| Size | 19.504 | 26.110 | 22.095 | 1.313 |
| ALR | 0.049 | 0.973 | 0.425 | 0.216 |
| TobinQA | 0.867 | 9.817 | 2.090 | 1.449 |
| Cash | −0.364 | 0.263 | −0.019 | 0.106 |
| OC | 0.060 | 2.519 | 0.612 | 0.426 |
| BRIR | −0.753 | 8.751 | 0.421 | 1.182 |
| Board | 5.000 | 15.000 | 8.569 | 1.687 |
| KPMU | 0.000 | 1.000 | 0.159 | 0.365 |
| EPS | 0.000 | 0.705 | 0.026 | 0.097 |

### 4.2. Benchmark Regression Results

Table 3 presents the benchmark regression results, examining the impact of external evaluation of corporate social responsibility on corporate environmental investment. The regression analysis was conducted in two columns: column (1) without controlling for any variables, and column (2) after controlling for all variables. The results indicate that all coefficients in the regression model are significantly positive at the 1% level. Specifically, the coefficient of CSR is 0.137 in column (1) and 0.147 in column (2), suggesting that for every 1% increase in external evaluation of corporate social responsibility relative to competitors within the industry, the scale of corporate environmental investment expands by 13.7% and 14.7%, respectively. These findings support the hypothesis that actively undertaking social responsibility promotes the expansion of corporate environmental investment (Hypothesis 1a) and reject the alternative hypothesis (Hypothesis 1b).

**Table 3.** Regression results.

| Variables | 1 | 2 |
|---|---|---|
| | EPI | EPI |
| CSR | 0.137 *** | 0.147 *** |
| | (3.613) | (3.366) |
| Size | | 2.417 *** |
| | | (4.370) |
| ALR | | 24.252 *** |
| | | (8.270) |
| TobinQA | | −0.998 *** |
| | | (−3.362) |
| Cash | | −66.631 *** |
| | | (−12.845) |
| OC | | −7.341 *** |
| | | (−5.874) |
| BRIR | | −0.972 *** |
| | | (−3.112) |
| Board | | −0.778 ** |
| | | (−2.328) |
| KPMU | | 18.315 *** |
| | | (9.724) |
| EPS | | 56.830 *** |
| | | (6.902) |
| _cons | 3.521 | −48.041 *** |
| | (1.286) | (−4.126) |
| Industry | Yes | Yes |
| Year | Yes | Yes |
| Adj. R2 | 0.086 | 0.107 |
| N | 29,454 | 29,454 |

Note: *** and **, represent the significance levels of regression coefficients at 1% and 5%, respectively, with robust standard errors in parentheses.

The inclusion of control variables is also important. For instance, firm size is positively related to environmental investment, indicating that larger companies generally have the resources and willingness to invest more in environmental initiatives. The coefficient of financial leverage is significantly positive, suggesting that companies with higher leverage ratios have a greater inclination to invest in environmental initiatives compared to those with lower leverage ratios. However, in the overall sample, firm value, corporate cash flows, operational capacity, development capacity, and board size are all significantly negatively related to corporate environmental investment. This may be because companies with higher Tobin's Q values, cash flows, asset turnover ratios, and revenue growth rates, and larger boards, face fewer apparent environmental issues and lower environmental regulatory intensity, resulting in a lower willingness to proactively invest in environmental initiatives. Additionally, the coefficients of key pollution monitoring units and environmen-

tal subsidies are significantly positive, indicating that listed companies subject to stronger environmental regulations and receiving more government environmental subsidies show a greater intention to invest in environmental initiatives. Overall, these findings suggest that if a company performs well in terms of social responsibility, it is likely to have a higher scale of environmental investment. The regression results, along with the analysis of control variables, provide valuable insights into the relationship between external evaluation of corporate social responsibility and corporate environmental investment.

*4.3. Robustness Test Results*

Considering the potential impact of various measurements for core variables, we replaced both the dependent and independent variables and performed the regressions again. For the dependent variable, in line with reference [23], we adopted an alternative standardization approach, using revenue as the denominator, to measure the capitalization of environmental investment by companies. It combines greening fees and emission fees from the "management expenses" item in the income statement of listed companies with the primary regressor, total capitalized environmental expenditure. For the independent variable, drawing from studies [85,86], we selected 12 aspects from the "Basic Information Table of Corporate Social Responsibility Reports" in the GTJA database to reflect the external evaluation of corporate social responsibility, as is shown in Table 4. The dummy values of the 12 sub-indicators were then summed and standardized to obtain a measure of social responsibility evaluation. The regression results for the regression's changing dependent and independent variables are shown in Table 5. The column (1) results show that after changing the measurement method for the dependent variable (EPI'), the coefficient of external evaluation of corporate social responsibility (CSR) remains significant and positive, with a coefficient of 0.015. Moreover, after modifying the measurement method for the independent variable, the coefficient of external evaluation of corporate social responsibility (CSR') remains significant and positive (in column 2), with a coefficient of 5.259. According to the above robustness test, the positive relationship between external evaluation of corporate social responsibility and corporate environmental investment remains consistent with the earlier findings.

Additionally, we included provincial fixed effects to address potential endogeneity concerns arising from the reciprocal causal relationship between the dependent and explanatory variables. The regression results are displayed in column (3) of Table 5. Even after accounting for year, industry, and provincial fixed effects, the coefficient of external evaluation of corporate social responsibility (CSR) remains significant and positive, with a coefficient of 0.129. This indicates that, relative to other competitors in the industry, a 1% increase in the evaluation of CSR can lead to a 12.9% increase in the scale of environmental investment. The positive association between a favorable external evaluation of social responsibility and the proactive expansion of environmental investment by companies persists in alignment with previous findings.

**Table 4.** Change explanatory variable measurement description.

| Variable | Variable Symbol | Variable Description |
|---|---|---|
| Environmental protection investment | EPI | (Enterprise environmental protection investment/main business income) × 100 |
| Whether to refer to the GRI Sustainability Reporting Guidelines | GRI | Set dummy variables, report according to GRI Sustainability Reporting Guidelines, assign a value of 1, otherwise 0 |
| Whether to disclose shareholder rights protection | SHP | Set dummy variable, report disclosure shareholder equity protection assignment is 1, otherwise 0 |
| Whether to disclose creditor protection | CRP | Set dummy variable, report disclosure creditor protection assignment is 1, otherwise 0 |
| Whether to disclose the protection of employee rights and interests | SP | Set the dummy variable, the report discloses that the employee rights protection assignment is 1, otherwise 0 |
| Whether to disclose supplier rights protection | DP | Set dummy variable, report discloses that supplier equity protection assignment is 1, otherwise 0 |
| Whether to disclose customer and consumer rights protection | CUP | Set dummy variable, report discloses that customer and consumer rights protection assignment is 1, otherwise 0 |
| Whether to disclose the environment and sustainability | EP | Set dummy variables, report discloses that environment and sustainability assignment is 1, otherwise 0 |
| Whether to disclose public relations and social welfare undertakings | PR | Set dummy variables, report discloses that public relations and social good is assigned a value of 1, otherwise 0 |
| Whether to disclose the social responsibility system construction and improvement measures | SC | Set dummy variables, report discloses that social responsibility system construction and improvement measures is assigned a value of 1, otherwise 0 |
| Whether to disclose the content of production safety | WS | Set the dummy variable, report discloses that safety production content assignment is 1, otherwise 0 |
| Whether to disclose the shortcomings of the company | DF | Set the dummy variable, the report discloses the deficiencies of the company and assigns a value of 1, otherwise 0 |
| Disclosure intention | IMD | Set dummy variable, voluntary disclosure value is 1, should be disclosed, otherwise 0 |

(The "Corporate social responsibility" label spans all rows from GRI through IMD.)

To ensure the inclusion of important variables, we added two additional control variables, namely, the turnover rate of cash and cash equivalents, and analyst attention, to the original model that already included nine control variables. A new regression analysis was conducted, and the results are presented in column (4) of Table 5. After incorporating these two control variables, the coefficient of external evaluation of corporate social responsibility (CSR) remains significant and positive, with a coefficient of 0.124. This implies that, when taking additional control variables into account, there is a 12.4% increase in the scale of environmental investment corresponding to a 1% increase in the evaluation of corporate social responsibility (CSR). The positive impact of a higher external evaluation of social responsibility on encouraging companies to actively expand their environmental investment remains consistent with the earlier findings.

**Table 5.** Robustness test results.

| Variables | 1 | 2 | 3 | 4 |
|---|---|---|---|---|
| | **Changing the Core Variables** | | **Provincial Fixed Effect** | **Adding Control Variables** |
| | EPI′ | EPI | EPI | EPI |
| CSR | 0.015 ** (2.359) | | 0.129 *** (2.935) | 0.124 *** (2.786) |
| CSR′ | | 5.259 ** (2.446) | | |
| Control variable | Yes | Yes | Yes | Yes |
| Cash turnover | | | | −0.061 (−0.953) |
| Ana attention | | | | 0.195 *** (2.709) |
| _cons | 11.287 *** (6.059) | −51.663 *** (−4.467) | −66.258 *** (−5.516) | −29.964 ** (−2.301) |
| Year | Yes | Yes | Yes | Yes |
| Industry | Yes | Yes | Yes | Yes |
| Province | | | Yes | |
| Adj. R2 | 0.098 | 0.107 | 0.111 | 0.107 |
| N | 29,454 | 29,454 | 29,454 | 29,454 |

Note: *** and **, represent the significance levels of regression coefficients at 1% and 5%, respectively, with robust standard errors in parentheses.

### 4.4. Discussion on Endogeneity Issues

We considered three aspects of endogeneity issues.

Firstly, to mitigate the potential problems related to omitted variables, we utilized a panel data fixed-effects model in this study. The regression results are displayed in Table 6. In column (1), it can be observed that the coefficient of external evaluation of corporate social responsibility (CSR) remains significant and positive, with a coefficient of 0.121. This indicates that, relative to industry competitors, a 1% increase in the evaluation of CSR can result in a 12.1% increase in the scale of environmental investment. The positive relationship between a higher external evaluation of social responsibility and the active expansion of environmental investment by companies continues to align with the earlier findings. (In untabulated results, we also explored measures of two potentially omitted variables that may influence corporate environmental investments: management incentive and corporate culture. Upon including these two variables in our regression model, our main results remained robust, indicating that our model is resilient to the inclusion of additional confounding factors. Details are available upon request, and we thank an anonymous referee for these suggestions).

Secondly, to address potential sample selection issues, this study employed the propensity score matching (PSM) technique. Using the median level of corporate social responsibility (CSR) performance as a basis, a dummy variable called CSRHL was created. Companies with CSR performance greater than the sample median were assigned a value of 1 as the treatment group, while companies with CSR performance lower than the sample median were assigned a value of 0 as the control group. Covariates, including the control variables utilized in this study, were employed for matching. The average treatment effect on the treated (ATT) was then calculated, resulting in a significant value of 2.01. Subsequently, a regression analysis was conducted using the matched sample. The results, presented in Table 6, column (2), indicate that the coefficient of external evaluation of CSR remains significant and positive, with a coefficient of 0.149. This suggests that, relative to industry competitors, a 1% increase in the evaluation of CSR can lead to a 14.9% increase in the scale of environmental investment. The positive impact of a higher external evaluation of social responsibility on encouraging companies to actively expand their environmental investment remains consistent with the earlier findings.

**Table 6.** Endogeneity test results.

| Variables | 1 | 2 | 3 | 4 |
|---|---|---|---|---|
| | **Panel Data Fixation Effect** | **PSM** | **Delayed Regression of EPI and CSR** | |
| | **EPI** | **EPI** | **EPI** | **EPI** |
| CSR | 0.121 *** (3.200) | 0.149 *** (0.055) | | |
| LCSR | | | 0.084 * (1.881) | |
| L2CSR | | | | 0.100 ** (2.280) |
| Control variable | Yes | Yes | Yes | Yes |
| _cons | −107.158 *** (−5.360) | 71.604 *** (8.740) | −48.931 *** (−3.903) | −53.642 *** (−4.083) |
| Year | Yes | Yes | Yes | Yes |
| Industry | Yes | Yes | Yes | Yes |
| Adj. R2 | 0.107 | 0.108 | 0.111 | 0.117 |
| N | 29,454 | 29,454 | 25,716 | 22,274 |

Note: ***, **, and * represent the significance levels of regression coefficients at 1%, 5%, and 10%, respectively, with robust standard errors in parentheses.

Thirdly, considering the possibility of reverse causality, when a company invests more in environmental protection, it may enjoy a higher CSR evaluation. Therefore, we regressed the lagged core explanatory variable one period and two periods in arrears to weaken the impact of reverse causality. The regression results, shown in Table 6, columns (3) and (4), for one-period lag and two-period lag, respectively, indicate that the coefficient of delayed CSR remains significant and positive, which aligns with the benchmark regression findings.

*4.5. Mechanism Analysis Results*

Based on the theoretical analysis and benchmark regression results above, the logic of shareholder value theory is supported. The fulfillment of corporate social responsibility (CSR) has several positive outcomes, including attracting positive media attention, building a good public image, attracting responsible consumers, and gaining the trust of suppliers and customers. These factors contribute to differentiation strategies, increased sales revenue, and improved product profitability. Additionally, fulfilling CSR can attract attention from external investors, facilitating access to financial resources from socially responsible investors and improving financing convenience for environmental investment. Building upon these findings, we recognize the importance of exploring the mechanisms through which CSR disclosure affects environmental investment.

For the information mechanism, we suppose that companies receiving higher evaluations for their social responsibility actively disclose information about their social responsibility actions to the public. As a result, these companies can establish a responsible image and a positive social reputation. Additionally, it improves the company's transparency, conveying a strong intention for sound management and long-term development to external stakeholders. Thus, we used corporate reputation (CR) and the proportion of institutional investors' shareholding (IIP) as proxy variables to investigate the mediating effects of this mechanism. According to the theory of corporate resources, corporate reputation is a valuable intangible asset that is scarce and difficult to imitate and substitute. We built this variable as referenced in [50]. Moreover, a positive evaluation of social responsibility attracts institutional investors who prioritize asset security and stable operational funds, increasing their ownership ratio. We built this variable as referenced in [51]. The establishment of such information transparency would create a "back-pressure" mechanism that puts pressure on the company itself, prompting it to actively expand its investments in green development and environmental protection to protect its good reputation and image.

Table 7 shows the mediating effect results. Columns (1) and (2) examine the mediating effect of corporate reputation (CR). In column (1), the coefficient of corporate social responsibility (CSR) is significantly positive, with a coefficient of 0.016 at the 1% level, indicating that fulfilling social responsibility can help companies establish a good social reputation.

In column (2), the coefficients of both social responsibility (CSR) and corporate reputation (CR) are significantly positive at the 10% and 1% confidence levels, respectively, demonstrating that corporate reputation plays a partially mediating role in the impact of external evaluation of social responsibility on corporate environmental investment. This suggests that a good corporate reputation is beneficial for prompting companies to actively expand their environmental investment scale. Columns (3) and (4) examine the mediating effect of institutional investor ownership (IIP). In column (3), the coefficient of external evaluation of corporate social responsibility (CSR) is significantly positive, with a coefficient of 0.079 at the 1% level, indicating that fulfilling social responsibility enhances the ownership ratio of institutional investors. In column (4), the coefficients of both external evaluation of social responsibility (CSR) and institutional investor ownership ratio (IIP) are significantly positive at the 1% and 10% confidence levels, respectively. This confirms that corporate reputation plays a partially mediating role in the impact of external evaluation of social responsibility on corporate environmental investment.

**Table 7.** Mediation effect of information transmission.

| Variables | 1 | 2 | 3 | 4 |
|---|---|---|---|---|
| | CR | EPI | IIP | EPI |
| CSR | 0.016 *** | 0.093 * | 0.079 *** | 0.142 *** |
| | (21.683) | (1.833) | (8.741) | (3.263) |
| CR | | 1.499 *** | | |
| | | (3.944) | | |
| IIP | | | | 0.045 * |
| | | | | (1.901) |
| Control variable | Yes | Yes | Yes | Yes |
| _cons | −40.422 *** | 18.147 | −158.530 *** | −41.263 *** |
| | (−158.171) | (0.937) | (−54.918) | (−3.395) |
| Year | Yes | Yes | Yes | Yes |
| Industry | Yes | Yes | Yes | Yes |
| Adj. R2 | 0.762 | 0.113 | 0.287 | 0.107 |
| N | 29,454 | 29,454 | 29,454 | 29,454 |

Note: ***, and * represent the significance levels of regression coefficients at 1%, and 10%, respectively, with robust standard errors in parentheses.

For the resource acquisition mechanism, the positive external evaluation of social responsibility indicates that the company values communication with stakeholders and conveys a signal to society that it prioritizes long-term goals and avoids short-term opportunistic behavior. This strengthens the company's governance structure and operational capabilities, leading to an increase in cash flows generated from business activities. It also reduces information asymmetry between the company and investors, alleviates financing constraints, and enhances the company's capacity to expand its environmental investment. Therefore, examining the mediating effect through the use of operating cash flow (OCF) and financing constraints (WW) as proxy variables for resource acquisition mechanisms can provide evidence for these relationships. Drawing on resource-based theory and stakeholder theory, a positive external evaluation of social responsibility promotes the generation of more abundant operating cash flow from business activities, thereby enhancing the company's capacity to expand environmental investment. Moreover, existing literature suggests that a higher external evaluation of social responsibility can increase transparency, reduce information asymmetry between companies and stakeholders, decrease external financing costs, alleviate financing constraints, and enhance the overall resources available to companies. The above mechanisms would increase the resources that can be used by companies, thus promoting proactive expansion of environmental investment.

Table 8 shows the mediating effect results. Columns (1) and (2) examine the mediating effect of operating cash flow (OCF). In column (1), the coefficient of corporate social responsibility (CSR) is significant and positive, with a coefficient of 0.001, indicating that

fulfilling social responsibility can lead to more abundant cash flow in the company's operations. In column (2), the coefficients of both social responsibility external evaluation (CSR) and operating cash flow (OCF) are significant and positive at the 5% and 1% confidence levels, respectively, suggesting that operating cash flow plays a partial mediating role in the impact of corporate social responsibility evaluation on environmental investment. In column (3), the coefficient of external evaluation of corporate social responsibility (CSR) is significant and negative, with a coefficient of −0.001, indicating that fulfilling social responsibility can effectively alleviate the financing constraints faced by the company. In column (4), the coefficient of external evaluation of social responsibility (CSR) is significant and positive at the 1% confidence level, demonstrating that financing constraints play a partial mediating role in the impact of external evaluation of corporate social responsibility on environmental investment. (We thank an anonymous referee for pointing out two other possible channels of mediation: signaling effects and compliance pressure. We used analyst attention and whether the company passed the ISO9001 [87] certification as proxies for these two channels. Results from the unreported tests also support these two mediating mechanisms for the effect of CSR on environmental investments to channel through. These additional test results are available upon request.)

**Table 8.** Mediation effect of resource acquisition.

| Variables | 1 | 2 | 3 | 4 |
|---|---|---|---|---|
|  | OCF | EPI | WW | EPI |
| CSR | 0.001 *** | 0.114 ** | −0.001 *** | 0.151 *** |
|  | (27.350) | (2.575) | (−40.904) | (3.121) |
| OCF |  | 48.265 *** |  |  |
|  |  | (6.072) |  |  |
| WW |  |  |  | −38.707 ** |
|  |  |  |  | (−2.321) |
| Control variable | Yes | Yes | Yes | Yes |
| _cons | −0.057 *** | −45.294 *** | 0.096 *** | −33.248 ** |
|  | (−6.864) | (−3.893) | (16.607) | (−2.442) |
| Year | Yes | Yes | Yes | Yes |
| Industry | Yes | Yes | Yes | Yes |
| Adj. R2 | 0.369 | 0.108 | 0.786 | 0.108 |
| N | 29,454 | 29,454 | 29,454 | 29,454 |

Note: *** and **, represent the significance levels of regression coefficients at 1% and 5%, respectively, with robust standard errors in parentheses.

### 4.6. Heterogeneity Analysis Results

4.6.1. Group Testing of State-Owned Enterprises and Non-State-Owned Enterprises

Differences in the expectations of corporate social responsibility (CSR) fulfillment may exist among enterprises with different ownership structures, which can impact the relationship between the external evaluation of CSR and the scale of environmental investment. State-owned enterprises, as important tools for macroeconomic regulation by the government, may have a higher initial burden of social responsibility compared to non-state-owned enterprises. To implement national development strategies, industrial policies, employment goals, and poverty alleviation plans, state-owned enterprises may prioritize non-economic activities and assume more social responsibility, even at the expense of some economic interests. This can lead to a higher external evaluation of social responsibility for state-owned enterprises. Therefore, this study expected that the external evaluation of CSR would not have a promoting effect on environmental investment for state-owned enterprises. The results of the segmented regression analysis are presented in Table 9. In column (1), which displays the regression results for state-owned enterprises, the coefficient of external evaluation of corporate social responsibility (CSR) is positive but not significant. In column (2), which presents the regression results for non-state-owned enterprises, the coefficient of external evaluation of corporate social responsibility (CSR) is significant and

positive, with a coefficient of 0.206. This means that for non-state-owned enterprises, for every 1% increase in the evaluation of CSR, the scale of environmental investment can increase by 20.6% relative to state-owned enterprises. This indicates that CSR has a greater promoting effect on environmental investment for non-state-owned enterprises, which aligns with expectations.

4.6.2. Group Testing of High-Level and Low-Level Environmental Subsidies

The level of environmental subsidies received by a company often has an impact on the level of environmental investment (EPI). The external evaluation of corporate social responsibility (CSR) can help companies establish stronger trust with the government, build stronger political connections, and obtain more favorable policy benefits, including support for environmental subsidies. In this study, the median grouping method was used to categorize companies based on the level of environmental subsidies received. Companies that receive environmental subsidies below the median were classified as the low environmental subsidy group, while companies that receive environmental subsidies above the median were classified as the high environmental subsidy group. The results of the segmented regression analysis are presented in Table 9. In column (3), which displays the regression results for companies in the low environmental subsidy group, the coefficient of external evaluation of corporate social responsibility (CSR) is significant and positive, with a coefficient of 0.193. This means that for companies in the low environmental subsidy group, for every 1% increase in the evaluation of CSR, the scale of environmental investment can increase by 19.3% relative to the high environmental subsidy group. In column (4), which presents the regression results for companies in the high environmental subsidy group, the coefficient of external evaluation of corporate social responsibility (CSR) is positive but not significant. This indicates that the promoting effect of CSR on environmental investment (EPI) is greater for companies in the low environmental subsidy group.

4.6.3. Group Testing of High-Level and Low-Level Environmental Philosophy Disclosure

The disclosure of a company's environmental philosophy, environmental policy, environmental management structure, circular economy development model, and green development in its annual report can have an impact on the level of environmental investment (EPI). When companies more comprehensively express these concepts, both internally and externally, they tend to have a stronger willingness to invest in environmental protection. In this study, A-share listed companies were grouped and tested based on whether they disclose the aforementioned information in their annual reports. Companies that disclose the information were assigned a value of 1, while those that do not disclose it were assigned a value of 0. The results of the segmented regression analysis are presented in Table 9. In column (5), which displays the regression results for companies that disclose the above information in their annual reports, the coefficient of external evaluation of corporate social responsibility (CSR) is significant and positive, with a coefficient of 0.247. This means that for companies that disclose the above information, for every 1% increase in the evaluation of CSR, the scale of environmental investment can increase by 24.7% relative to companies that do not disclose the above information. In column (6), which presents the regression results for companies that do not disclose the above information in their annual reports, the coefficient of external evaluation of corporate social responsibility (CSR) is positive but not significant. This indicates that the promoting effect of CSR on environmental investment (EPI) is greater for companies that disclose the above information in their annual reports.

4.6.4. Group Testing of High-Level and Low-Level Environmental Philosophy Disclosure

Whether a company is identified as a key pollution-monitoring unit can have an impact on the effectiveness of implementing external evaluation of corporate social responsibility (CSR). When a company is disclosed as a key pollution-monitoring unit in reports, it tends

to be more proactive in assuming social responsibility to gain a social reputation and access to high-quality resources. As a result, these companies actively expand their scale of environmental investment (EPI). To investigate this further, this study examined whether the impact of CSR on environmental investment (EPI) differs between companies disclosed as key pollution-monitoring units in reports and those not disclosed as such. Companies disclosed as key pollution-monitoring units in reports were assigned a value of 1, while companies not disclosed as key pollution monitoring units were assigned a value of 0. The results of the segmented regression analysis are presented in Table 9. In column (7), which displays the regression results for companies disclosed as key pollution-monitoring units in reports, the coefficient of CSR is significant and positive, with a coefficient of 0.497. This means that for companies disclosed as key pollution-monitoring units, for every 1% increase in CSR, the scale of environmental investment can increase by 49.7% relative to companies not disclosed as key pollution-monitoring units. In column (8), which presents the regression results for companies not disclosed as key pollution-monitoring units in reports, the coefficient of CSR is also significant and positive, with a coefficient of 0.102. This means that for every 1% increase in CSR, the scale of environmental investment can increase by 10.2% for these companies. However, the results of the inter-group difference test show Prob > chi2 = 0.0185, indicating that the promoting effect of CSR on environmental investment (EPI) is greater for companies disclosed as key pollution-monitoring units in reports.

**Table 9.** Heterogeneity analysis results.

| Variable | State-Owned Enterprise | Non-State-Owned Enterprise | Low Environmental Grant Group | High Environmental Grant Group | Environmental Philosophy Disclosed | Environmental Philosophy Not Disclosed | Key Pollution-Monitoring Unit | Not Key Pollution-Monitoring Unit |
|---|---|---|---|---|---|---|---|---|
| | 1 | 2 | 3 | 4 | 5 | 6 | 7 | 8 |
| | EPI | EPI | EPI | EPI | EPI | EPI | EPI | EPI |
| CSR | 0.096 | 0.206 *** | 0.193 *** | 0.103 | 0.247 *** | 0.045 | 0.497 *** | 0.102 ** |
| | (1.530) | (3.261) | (3.290) | (1.620) | (2.954) | (0.887) | (3.153) | (2.227) |
| Control variable | Yes | Yes | Yes | Yes | Yes | Yes | Yes | Yes |
| _cons | 10.491 | −89.823 *** | −20.326 | −71.306 *** | −14.281 | −83.351 *** | −3.680 | −56.601 *** |
| | (0.572) | (−5.171) | (−1.435) | (−3.816) | (−0.688) | (−5.369) | (−0.107) | (−4.690) |
| Year | Yes | Yes | Yes | Yes | Yes | Yes | Yes | Yes |
| Industry | Yes | Yes | Yes | Yes | Yes | Yes | Yes | Yes |
| Adj. R2 | 0.172 | 0.071 | 0.078 | 0.119 | 0.141 | 0.084 | 0.171 | 0.076 |
| N | 11,189 | 18,265 | 14,755 | 14,699 | 8877 | 20,577 | 4768 | 24,686 |

Note: *** and **, represent the significance levels of regression coefficients at 1% and 5%, respectively, with robust standard errors in parentheses.

### 4.7. Additional Analysis: ESG

A considerable volume of literature exists that positions ESG (environmental, social, and governance) as an alternative metric for assessing corporate social responsibility. Recognized as a pivotal indicator within the spectrum of social responsibility, ESG shares a discernible correlation with CSR. Such interconnection in our findings beckons additional scrutiny. In our analysis, we contemplated the principles of both CSR and ESG, alongside their developmental trajectories in the Chinese milieu. This reflective approach ensures a comprehensive understanding of these frameworks as they adapt and transform within China's unique corporate landscape.

Bowen's seminal proposition of corporate social responsibility (CSR) in 1953 set the foundation for recognizing that companies should extend their concerns beyond profit to encompass the interests of various stakeholders. He posited that economic gains should not overshadow a company's social duties—a principle that has captivated scholarly attention and remains a focal point of research. To date, the exploration of CSR's influences persists robustly. For example, Refs. [88,89] delve into the informational aspects of social responsibility disclosures, while studies by Refs. [90–92] assess how social responsibility affects firm performance and financial constraints. Additionally, the intersections of CSR disclosure and green innovation are being investigated, illustrating the concept's broadening

scope [93,94]. ESG theory has its roots in the concept of responsible investment. Initially, it questioned the potential of fund allocation to indirectly steer corporate decisions toward mitigating negative outcomes associated with racial tensions, environmental degradation, and warfare. These selective investments laid the groundwork for what would become known as ESG investing. The 2006 launch of the United Nations Principles for Responsible Investment by the New York Stock Exchange marked a pivotal moment, fostering the convergence of social responsibility issues on a global platform. By 2017, the initiative had garnered the support of more than 1750 investment institutions—overseeing a collective sum surpassing USD 70 trillion—and was pivotal in weaving ESG considerations into the fabric of investment analysis and decision-making processes [95]. The evolution of these principles has notably heightened investor awareness of corporate social responsibility, shaping the landscape of modern investment practices.

In the backdrop of China's late-20th-century economic expansion, a pursuit of profit by many enterprises resulted in adverse environmental and societal ramifications. This triggered widespread public disapproval, prompting the government to enact robust environmental protection laws and compel social responsibility disclosures from industries deemed heavy polluters. Although such mandatory disclosures were not extended to all business types, the span from 2011 to 2020 witnessed a marked increase in social responsibility reports within the Chinese A-share market, soaring from 565 to 1005 annual disclosures. This trajectory gained further momentum as China set forth ambitious targets for carbon peaking and carbon neutrality, accentuating the demand for sustainability-focused information. Pre-2020, the emphasis among listed Chinese companies was predominantly on the publication of CSR reports. However, the 2019 enhancement of the Environmental, Social, and Governance Reporting Guidelines—incorporating pivotal ESG metrics—catalyzed a transition towards the integration of social responsibility insights within ESG disclosures. This strategic shift is reflected in the growing prevalence of ESG ratings among A-share listed companies, underscoring a harmonization of CSR and ESG principles as cornerstones of China's commitment to sustainability.

Considering that both ESG and CSR reflect the level of social responsibility undertaken by companies, ESG places more emphasis on guiding investment institutions throughout the investment process, while CSR highlights the responsibility and obligations of companies towards society. They represent two different perspectives on the same concept. Existing research on ESG demonstrates the impact of ESG information. Evaluating social responsibility information effectively mitigates earnings management [96], ensures financial stability [97], and promotes innovation quality improvement. Studies show that the attention of investors and the general public plays a crucial role. This indicates that ESG, in terms of influencing investment decisions, also attracts the attention of stakeholders and generates reputational effects at the corporate level. Consequently, it encourages companies to engage in environmental investment to maintain their reputation and improve information transparency. Therefore, this study utilized companies' Bloomberg ESG ratings and Huazheng ESG ratings as external evaluation data for corporate social responsibility and conducted regression tests to verify the main hypothesis. The results, as shown in the Table 10, demonstrate a significant increase in the number of environmental investments by companies due to ESG ratings, further confirming the conclusions of our main results.

**Table 10.** Additional test results.

| Variables | 1 | 2 |
|---|---|---|
| | EPI | EPI |
| HuazhengESG | 0.589 *** | |
| | (5.465) | |
| BloombergESG | | 0.429 *** |
| | | (2.589) |
| Size | 2.406 *** | 0.326 |
| | (4.375) | (0.326) |
| ALR | 26.392 *** | 13.923 ** |
| | (8.686) | (2.424) |
| TobinQA | −0.807 *** | −1.080 * |
| | (−2.583) | (−1.871) |
| Cash | −66.072 *** | −102.956 *** |
| | (−12.244) | (−9.238) |
| OC | −7.381 *** | −5.378 ** |
| | (−5.697) | (−2.468) |
| BRIR | −0.923 *** | −0.783 |
| | (−2.759) | (−1.142) |
| Board | −0.840 ** | −0.442 |
| | (−2.472) | (−0.821) |
| KPMU | 17.617 *** | 10.907 *** |
| | (9.231) | (3.942) |
| EPS | 54.807 *** | 65.487 *** |
| | (6.505) | (4.204) |
| _cons | −85.357 *** | −5.084 |
| | (−6.823) | (−0.242) |
| Industry | Yes | Yes |
| Year | Yes | Yes |
| Adj. R2 | 0.110 | 0.145 |
| N | 28,308 | 9642 |

Note: ***, **, and * represent the significance levels of regression coefficients at 1%, 5%, and 10%, respectively, with robust standard errors in parentheses.

## 5. Policy and Managerial Suggestions

Our study provides important theoretical and practical implications. Firstly, in the case of theoretical implications, this study contributes to the research field of corporate social responsibility (CSR) by linking the external evaluation of CSR with environmental protection investment. It delineates the mechanisms through which CSR evaluation promotes companies to proactively expand their environmental investment, including information transmission and resource acquisition pathways. The study reveals the heterogeneous impacts of China's unique institutional background and current environment, enriching the existing literature in the field of CSR. Secondly, in the case of policymaking implications, this study offers valuable insights and empirical evidence to enhance the evaluation and investment in corporate social responsibility (CSR) and environmental protection among Chinese companies. It empirically investigates the relationship between CSR evaluation and environmental investment, examines the mechanisms through which CSR evaluation influences companies' environmental investment, and explores the varying impacts of property rights, environmental subsidies, and external supervision on this relationship. Drawing upon these findings, specific recommendations are provided, which hold practical significance for companies in actively fulfilling their social responsibility and expanding their environmental investments.

Based on the empirical findings of this study, we propose the following recommendations for both businesses and governments.

Firstly, we suggest that companies adopt a long-term strategic perspective and integrate environmental regulations into their goals to enhance their competitive advantage. Companies should assume more social responsibility and increase the level of disclosure

per relevant laws and regulations. This can be achieved through annual reports and specialized CSR disclosure reports, presenting their social responsibility initiatives to external stakeholders. By signaling and enhancing information transparency, companies can build a positive reputation and gain access to abundant resources, thus creating a back-pressure loop that strengthens the motivation for environmental investments.

Secondly, we recommend that governments consider the different types of companies when formulating environmental promotion policies. Targeted environmental regulations should be issued, with a particular focus on providing resources and support to private enterprises. The government should play a macroeconomic role by increasing funding and technological assistance, effectively promoting environmental actions among numerous private enterprises. It is crucial to further refine the classification and evaluation of key polluting enterprises, strengthen regulatory oversight, and facilitate the transition of highly polluting companies towards cleaner production.

Lastly, the government should introduce policies that encourage companies to disclose their social responsibility information more comprehensively. Channels for accessing such information in the capital market should be improved. By guiding public attention towards corporate social responsibility information and promoting the concept of sustainable development across society, a mechanism of constraint can be established to encourage companies to fulfill their environmental protection role effectively.

## 6. Conclusions

This paper examines the relationship between the external evaluation of CSR performance and environmental protection investment behavior of China's A-share listed companies from 2010 to 2021 from the perspective of shareholder value.

Through all the empirical analyses, we have several meaningful conclusions. We find that the disclosure of CSR information, as identified by third-party rating agencies, plays a significant role in promoting enterprises to actively improve their environmental protection investment. This finding remained robust even after conducting various tests to address potential biases, indicating that high-quality CSR disclosure can effectively facilitate environmentally friendly resource allocation, promote overall sustainable development, and contribute to the theory of shareholder value. Then, based on the baseline regression, this study established two mediating mechanisms, namely, the information transmission mechanism and the resource acquisition mechanism, to explain the underlying reasons for the observed relationship. The quality of CSR information disclosure enhances information transparency through reputation and the involvement of professional investment institutions. This prompts companies to strengthen their environmental protection investment to maintain a positive external image. Additionally, sufficient cash flow and reduced financing constraints enable companies to access more resources, thus reducing limitations on environmental protection investment and fostering greater environmental commitment. At last, heterogeneity analysis revealed that the relationship between CSR disclosure and environmental protection investment is more pronounced in non-state-owned enterprises, companies receiving fewer environmental protection subsidies, companies that disclose environmental protection concepts, and companies that highlight their role as key pollution-monitoring units in their reports. These findings further elucidate the mechanisms through which CSR information disclosure effectively operates.

In summary, against the backdrop of China's efforts to promote green and sustainable development, these research findings emphasize the continued importance of Chinese enterprises disclosing CSR information and optimizing the quality of third-party rating agencies. Such initiatives will effectively drive companies to reduce pollution and carbon emissions through environmental protection investment, enhance the quality of green innovation, improve their overall sustainable development capabilities, and contribute to China's goals of achieving a carbon peak and optimizing the ecological environment.

There are some limitations to this research. The voluntary nature of corporate social responsibility (CSR) disclosure may lead to incomplete disclosure of CSR information

within the sample, potentially impacting the conclusions to some extent. The disclosure of corporate information is significantly influenced by managerial characteristics, such as gender, education level, and experiences of CEOs. These managerial traits, in turn, have an impact on companies' investment decisions. These aspects present opportunities for future research, such as refining the collection of CSR disclosure information and investigating whether CSR's impact on environmental investment varies under the influence of different managerial traits.

**Author Contributions:** Conceptualization, R.L. and M.W.; data curation, F.S.; formal analysis, F.S. and Y.Z.; methodology, R.L. and Y.Z.; software, F.S.; supervision, M.W. and Y.Z.; writing—original draft, R.L. and F.S.; writing—review and editing, R.L. All authors have read and agreed to the published version of the manuscript.

**Funding:** This research received no funding.

**Institutional Review Board Statement:** Not applicable.

**Informed Consent Statement:** Not applicable.

**Data Availability Statement:** The data that support the findings of this study are available on request from the corresponding author. The data are not publicly available due to privacy or ethical restrictions.

**Conflicts of Interest:** The authors declare no conflicts of interest. There is no professional or other personal interest of any nature or kind in any product, service, and/or company that could be construed as influencing the position presented in, or the review of, the manuscript entitled.

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
