# Peer review of "The Impact of Corporate Social Responsibility on Environmental Investment: The Mediating Effects of Information Transmission and Resource Acquisition"

_sustainability, doi:10.3390/su16062457_

Round 1

Reviewer 1 Report

Comments and Suggestions for Authors

The article is interesting, The examples are well chosen. Complies with the requirements

Reviewer 2 Report

Comments and Suggestions for Authors

1. Methodological Limitations

Sample Selection and Scope: The study mainly focuses on Chinese A-share listed companies, which may limit the generalizability of its results. Similar research on companies from different countries or regions may reveal different dynamics or additional influencing factors.

Endogeneity Issues: Despite using methods like Propensity Score Matching (PSM) to address potential endogeneity issues, there may still be omitted unobserved variables that could affect the relationship between CSR and environmental investment. Factors such as management incentives and corporate culture may not have been fully considered.

2. Data and Variable Definitions

Data Accessibility and Transparency: The study relies on publicly available data and third-party ratings, which may suffer from information disclosure inadequacies, affecting the accuracy and depth of the results.

Operationalization of Variables: The definitions and measurements of CSR and environmental investment may be subjective, and different operationalization methods may lead to varying results.

3. Theoretical Framework and Hypothesis Testing

Scope of Theoretical Assumptions: While assumptions are built around "Shareholder Value Theory" and "Managerial Opportunism Theory," other theoretical perspectives (such as Stakeholder Theory) might not have been fully leveraged to explain the complex relationship between CSR and environmental investment.

Completeness of Mediating Mechanisms: While information dissemination and resource acquisition are mentioned as mediating mechanisms, other possible mechanisms (such as signaling effects, compliance pressures, etc.) may not have been fully considered.

4. Interpretation and Discussion of Empirical Results

Generalizability of Conclusions: The study's results are based on data from a specific period, and over time, the impact of CSR on environmental investment may vary.

Practical Implications of Research Results: While the study provides insights into the relationship between CSR and environmental investment, further exploration may be needed on how to effectively utilize these findings in guiding corporate strategies and policy-making.

Conclusion

The study offers valuable insights and meaningful contributions to the relevant literature. However, identified potential drawbacks and areas for improvement suggest directions for future work, including expanding the geographic and industry scope of the research, delving deeper into the complexity of endogeneity and mediating mechanisms, and enhancing the practicality and applicability of research results. These improvements not only strengthen the robustness of existing conclusions but also further expand the knowledge boundaries of theoretical and practical domains.

Reviewer 3 Report

Comments and Suggestions for Authors

In this study, the authors conducted a regression analysis on the external evaluation of CSR and the environmental investments of Chinese listed companies. The paper has an acceptable structure. The authors carry out empirical research, and the logic of obtaining the results is respected. The obtained results are interpreted and highlighted in the paper Some aspects need checking:
1.  Line 335/352. I recommend integrating the bolded elements into the text, without marking them as subtitles. You already mentioned examining the role of various mediating variables to validate those mechanisms.
2. In section 4.2.2. Robustness Test Results,  the authors comment based on Table 5. This Table is presented much later. I recommend restructuring the content or reordering the tables. Also, I recommend integrating the numbering/bolded elements into the text, without marking them as subtitles.
3. Sections 4.2.4. Mechanism Analysis Results / 4.2.5. Heterogeneity Analysis Results. The same recommendation to integrate numbering/bolded elements into the text, without marking them as subtitles.
4. Line 534. This sentence must be improved.
5. It is not necessary to divide the Conclusions section into subsections.

Reviewer 4 Report

Comments and Suggestions for Authors

Journal: Sustainability (ISSN 2071-1050)

Manuscript ID: Sustainability-2876014

Type : Article

Title :The Impact of Corporate Social Responsibility on Environmental Investment: The Mediating Effects of Information Transmission and Resource Acquisition 

Thank you for the opportunity to review your work. In order to investigate the impacts of corporate social responsibility (CSR) on business activities and study the empirical research on the relationship between CSR and environmental investment, the authors conduct a regression analysis on the external evaluation of CSR and enterprises' environmental investment using data from Chinese listed companies. I think that this paper has a merit in terms of the considered problem setting. I suggest that a minor revision be made to the paper before I consider it as competent for publication. Some minor revision are listed as follows:

First, lines 44-46, ‘In the long term, according to the Porter Hypothesis, enterprises that increase their environmental investment can effectively stimulate the innovation and adoption of clean technologies. ‘ The authors need to add the reference of ‘Porter Hypothesis’ and other definitions for the first time in the paper.

Second, lines 88-89, ‘Corporate social responsibility plays a crucial role in aligning economic and social 88 development, ecological civilization construction, and social harmony and stability.’ The authors need to give the definition of ‘corporate social responsibility (CSR)’ for the first time in the paper, which should not give the definition in line 170, and use CSR for the following parts of the paper, for example, line 134.

Third, lines 158-160, ‘In 2020, Chinese leaders announced China's commitment to peak CO2 emissions by 2030 and achieve carbon neutrality by 2060 during the 75th United Nations General Assembly. ‘The authors need to add the support reference of this data and other data in the paper. 

Fourth, in the Introduction Section, the theoretical contribution of this paper is needed to be discussed, which are neither distinct nor convincing as far as I'm concerned.

Comments on the Quality of English Language

no comments

Reviewer 5 Report

Comments and Suggestions for Authors

You have the potential to conduct and publish an empirical paper on environmental investment and organizational strategies in this regard. However, your focus on CSR is narrow and outdated. CSR's impact on corporate performance and social development has been widely researched and published. I strongly recommend you follow the similar logic but focus on ESG instead, which is the emerging and under-researched area.

Comments on the Quality of English Language

Your writing is well-structured and written, with a few minor issues only.

Round 2

Reviewer 2 Report

Comments and Suggestions for Authors

Accept in present form
